

# Towards a low-resolution infrared sounder for monitoring atmospheric ammonia (NH$_3$) at high spatial resolution

Lara Noppen[1], Lieven Clarisse[1], Frederik Tack[2], Thomas Ruhtz[3], Martin Van Damme[1,2], Michel Van Roozendael[2], Dirk Schuettemeyer[4], and Pierre Coheur[1]

[1]Spectroscopy, Quantum Chemistry and Atmospheric Remote Sensing (SQUARES), BLU-ULB Research Center, Université libre de Bruxelles (ULB), Brussels, Belgium
[2]Royal Belgian Institute for Space Aeronomy (BIRA-IASB), Brussels, Belgium
[3]Department of Earth Sciences, Freie Universität Berlin, Berlin, Germany
[4]European Space Agency (ESA-ESTEC), Noordwijk, the Netherlands

**Correspondence:** Lara Noppen (lara.noppen@ulb.be)

**Abstract.** Over the past decade, hyperspectral infrared sounders on satellites have offered global measurements of atmospheric ammonia (NH$_3$), providing valuable insights into its sources. However, due to their coarse spatial resolution and gaps in spatial coverage, inferring emissions from smaller sources or utilizing data from single overpasses remains very challenging. While a high spatial resolution imaging-sounder would greatly enhance monitoring capabilities, developing an instrument that

combines high spatial and spectral resolution is technologically difficult and expensive. Here, we analyze the feasibility of measuring NH$_3$ with instruments having a largely reduced spectral coverage and resolution compared to current operational sounders. We explore the performance trade-offs using simulated spectra, measurements from the Infrared Atmospheric Sounding Interferometer (IASI) satellite sounder, and spectra obtained from aircraft. The measured spectra are degraded spectrally, and their performance is evaluated using metrics such as NH$_3$ measurement uncertainty, signal-to-noise ratio, and false alarm

rate. Instruments that measure across a continuous spectral interval and instruments covering specific well-chosen spectral bands are both examined. We demonstrate that a future dedicated NH$_3$ sounder with as few as three spectral bands of 1–5 cm$^{-1}$ is feasible and would enable the detection of NH$_3$ at both high spatial resolution and across continental scales. The advantage of choosing well-defined spectral bands is demonstrated, e.g. by showing that an instrument with five specific bands of 5 cm$^{-1}$ performs similarly to one with 20 contiguous channels across 900–1000 cm$^{-1}$. Additionally, we show that at high

spectral resolutions (below 5 cm$^{-1}$), the NH$_3$ measurement capability is primarily driven by the instrumental noise. As the spectral resolution or number of measurement bands decreases, spectral interferences from other atmospheric constituents and the surface start to dominate the NH$_3$ retrieval uncertainty budget, fundamentally limiting the unambiguous identification of NH$_3$.

## 1  Introduction

Atmospheric ammonia (NH$_3$) plays a key role in the formation of secondary particulate matter, which leads to chronic respiratory illnesses and premature deaths (Apte et al., 2018; Wyer et al., 2022). Excess NH$_3$ also poses a significant threat to the



environment, particularly through the acidification and eutrophication of ecosystems, leading to a loss of biodiversity, degradation of water and soil quality, and other environmental issues (de Vries et al., 2024). The primary source of excess atmospheric $NH_3$ is intensive agricultural activities, including livestock farming and the use of synthetic fertilizers (Sutton et al., 2013).

Global measurements of $NH_3$ are currently provided by satellites equipped with hyperspectral infrared sounders, such as the Infrared Atmospheric Sounding Interferometer (IASI, Clarisse et al. (2009)) and the Cross-track Infrared Sounder (CrIS, Shephard and Cady-Pereira (2015)). Their measurements have provided valuable insights into the distribution and sources of $NH_3$ in the atmosphere. In particular, they revealed that strong point sources such as animal feedlots and industrial fertilizer production facilities dominate ambient $NH_3$ (Van Damme et al., 2018; Clarisse et al., 2019). However, their spatial resolution

(larger than 10 km) and non-contiguous coverage pose significant challenges for detecting smaller $NH_3$ emission sources and quantifying emissions from single overpasses. The agricultural landscape in Europe, for instance, features hundreds of thousand of farms that are too small or too agglomerated to be quantified individually with IASI (ESA, 2023). Even future instruments, such as the IASI-New Generation (IASI-NG, Crevoisier et al. (2014)) or the hyperspectral Infrared Sounder (IRS, Holmlund et al. (2021)), while offering considerable improvements, are not expected to provide the high spatial resolution necessary for

resolving small-scale point sources. In the context of ESA's Earth Explorer 11 call, a mission called Nitrosat (ESA, 2023) was proposed for measuring $NH_3$ and nitrogen dioxide ($NO_2$) at high spatial resolution. Within the Phase 0 studies, it was demonstrated that a dedicated sounder capable of measuring $NH_3$ at sub-kilometer resolution within a contiguous image 80 km across track is well within the capabilities of current technology. While the immense scientific and societal benefits of such a mission are unquestionable given the major disruption to the nitrogen cycle and the damaging consequences that this entails,

the proposed baseline instrument (characterized by a spectral range extending from 925 to 975 $cm^{-1}$ and a resolution of $0.625$ $cm^{-1}$) comes at a substantial cost.

    In this paper, we explore hypothetical spectrometers measuring infrared spectra at a lower spectral resolution or with a limited number of channels, and we investigate to what extent these can be used to effectively measure $NH_3$. By compromising on spectral resolution, imagers monitoring $NH_3$ at high spatial resolution could be developed, offering significant advantages in

terms of weight, size, cost, and development time. We begin with a brief description of the data used for this study, including IASI satellite measurements and Telops Hyper-Cam airborne data (Section 2). Following this, we discuss ways to quantify the $NH_3$ retrieval uncertainty. We also provide a concise overview of the matched filter technique used to detect and identify $NH_3$ features in infrared measurements, and the metrics used to assess the resulting distributions. In Section 4, we evaluate the performance of hypothetical instruments for measuring $NH_3$. We first analyze simulated spectra to characterize the effects of

spectral range and resolution. After that, a more comprehensive evaluation is performed using the remotely measured spectra. We evaluate the native performance of these instruments, and progressively downgrade their spectra by averaging channels, allowing the analysis of instruments with coarser spectral resolution. We investigate the performance not only for instruments with continuous spectral coverage and fixed sampling, but also for instruments measuring a limited number of specific spectral bands of various widths. In the final section, we present our conclusions.





## 2 Measurements


The spaceborne data used in this study were acquired by the IASI/MetopB instrument on 8 April 2020 (Figure 2). IASI is a passive sounder equipped with a Fourier-transform spectrometer, observing the Earth in the thermal infrared in nadir geometry. Operating in a sunsynchronous polar orbit, the satellite provides twice-daily global data, though only data from the morning overpass (around 9:30 AM local time) were used here. The instrument's field of view corresponds to a circular footprint of

12 $\mathrm{km}$ in diameter at nadir, extending elliptically up to $20 \times 39 \ \mathrm{km}^2$ for larger viewing angles. The spectral range of IASI covers 645 to 2760 $\mathrm{cm}^{-1}$, but was limited here to 812–1126 $\mathrm{cm}^{-1}$ following the range used for the IASI $\mathrm{NH_3}$ retrieval algorithm (Clarisse et al., 2023). The IASI L1C data have an apodized spectral resolution of 0.5 $\mathrm{cm}^{-1}$ and are sampled at 0.25 $\mathrm{cm}^{-1}$ (Clerbaux et al., 2009).

We also use measurements from aerial surveys that were carried out in Italy in the spring of 2022 in the context of Nitrosat
demonstration campaigns (ESA, 2023). The measurements were recorded by a Hyper-Cam LW on board a Cessna T207A aircraft. At an altitude of about 3 $\mathrm{km}$, the ground instantaneous field of view for each individual pixel was about 5 $\mathrm{m}$. The Hyper-Cam LW instrument, developed by Telops, is an advanced hyperspectral imaging camera employing Fourier-transform technology to deliver high-resolution spectra. Operating in the longwave infrared between 830 and 1270 $\mathrm{cm}^{-1}$, the spectrometer offers a spectral resolution ranging from 0.25 to 150 $\mathrm{cm}^{-1}$ (Lagueux et al., 2009a, b; Montembeault et al., 2010). Data from
two flights are exploited here. The first one (Figure 7) measured a scene of approximately 6 by 14 $\mathrm{km}$ centered on the soda ash production plant in Rosignano. The production of soda ash is based on the well-known Solvay process that notably requires an input of $\mathrm{NH_3}$. Although a large amount is recycled in the process, a part of the $\mathrm{NH_3}$ is lost to the atmosphere (European Commission, 2007). The $\mathrm{NH_3}$ emissions of the Rosignano plant were estimated to exceed 200 $\mathrm{t}$ in 2022 according to the European Pollutant Release and Transfer Register (E-PRTR, European Environment Agency (2022)). The area was surveyed on
19 May 2022, around noon. The second scene (Figure 8) was acquired on 30 June 2022 between 10 AM and 5 PM and covered an area of 10 by 55 $\mathrm{km}$ between the cities of Trecasali and Vetto. The northern part of the scene is part of the Po Valley, which is Europe's largest $\mathrm{NH_3}$ hotspot due to intensive agricultural activities (Lonati and Cernuschi, 2020; Van Damme et al., 2022). Between 2020 and 2022, the E-PRTR listed 8 agricultural $\mathrm{NH_3}$ emission sources in the surveyed area. In this study, for each of the two flights, the observed pixels were averaged two by two to reduce the noise (see Appendix A for the estimation of the
noise), resulting in a reduction of the spatial resolution from 5 to 10 $\mathrm{m}$. The native spectra are sampled at 1.3 $\mathrm{cm}^{-1}$.

## 3 Performance metrics and the matched filter

### 3.1 Measurement uncertainty

In nadir viewing geometry, the effect of atmospheric $\mathrm{NH_3}$ on observed radiances is linear for all but the largest columns (Clarisse et al., 2023). That is, the observed spectrum can be written as

$$\boldsymbol{y} = \boldsymbol{y_0} + x\boldsymbol{K} + \boldsymbol{\epsilon}, \tag{1}$$



with $\boldsymbol{y_0}$ the theoretical spectrum that would be observed in the absence of $NH_3$ and instrumental noise $\boldsymbol{\epsilon}$, $x$ the $NH_3$ total column and $\boldsymbol{K} = \frac{\partial \boldsymbol{y}}{\partial x}$ the Jacobian with respect to $NH_3$. If we assume that, except for $x$ and $\boldsymbol{\epsilon}$, we have perfect knowledge of all parameters affecting the spectrum (i.e. $\boldsymbol{y_0}$ and $\boldsymbol{K}$ are known), the weighted least-squares solution provides the following estimate for $x$ (Manolakis et al., 2016)

$$\hat{x} = \left( \boldsymbol{K}^T \mathbf{S}_\epsilon^{-1} \boldsymbol{K} \right)^{-1} \boldsymbol{K}^T \mathbf{S}_\epsilon^{-1} (\boldsymbol{y} - \boldsymbol{y_0}), \tag{2}$$

with $\mathbf{S}_\epsilon^{-1}$ the covariance matrix of the instrumental noise. This solution minimizes the (weighted) distance between the observed spectrum $\boldsymbol{y}$ and $\boldsymbol{y_0} + x\boldsymbol{K}$ in the presence of noise. The estimate $\hat{x}$ has an associated variance or measurement uncertainty (Bauduin et al., 2017)

$$\sigma_{\text{noise}} = \left( \boldsymbol{K}^T \mathbf{S}_\epsilon^{-1} \boldsymbol{K} \right)^{-1/2}, \tag{3}$$

which is the propagation of the instrumental noise to $\hat{x}$.

In practice, the state of the atmosphere or the surface is not known perfectly. However, we can still write

$$\boldsymbol{y} = \boldsymbol{y}_g + x\boldsymbol{K} + \boldsymbol{g}, \tag{4}$$

with now $\boldsymbol{y}_g$ some reference spectrum with negligible $NH_3$ and $\boldsymbol{g}$ a generalized noise vector, which includes both the contribution of the instrumental noise and spectral contributions related to the uncertain scene parameters. This includes, for example, unknown surface and air temperatures, or parameters that can directly interfere with the identification of the contribution of the $NH_3$ signature in the spectrum, such as unknown water vapor abundance and surface emissivity variations. If we can estimate the covariance matrix $\mathbf{S}_g$ associated with $\boldsymbol{y}_g$ and the Jacobian $\boldsymbol{K}$, we can, as before, obtain a weighted least-squares estimate (von Clarmann et al., 2001; Manolakis et al., 2016)

$$\hat{x} = \left( \boldsymbol{K}^T \mathbf{S}_g^{-1} \boldsymbol{K} \right)^{-1} \boldsymbol{K}^T \mathbf{S}_g^{-1} (\boldsymbol{y} - \boldsymbol{y}_g), \tag{5}$$

with uncertainty

$$\sigma_{\text{abs}} = \left( \boldsymbol{K}^T \mathbf{S}_g^{-1} \boldsymbol{K} \right)^{-1/2}. \tag{6}$$

This solution minimizes the distance between $\boldsymbol{y}$ and $\boldsymbol{y}_g + x\boldsymbol{K}$, but this time inversely weighted with a covariance matrix characterizing not only the instrumental noise but also spectral interferences. This least-squares solution can be shown to correspond to the maximum likelihood estimate in case $\boldsymbol{g}$ follows a multivariate normal distribution. Importantly, it can also be shown to be equivalent to the solution obtained by performing a simultaneous optimal estimation of the interfering spectral features (von Clarmann et al., 2001). In the field of hyperspectral imaging, $\boldsymbol{g}$ is referred to as the background clutter (Manolakis et al., 2014). The approach is very suitable for short-lived trace gases like $NH_3$, for which $\boldsymbol{y}_g$ and $\mathbf{S}_g$ can easily be estimated from observations with background levels of $NH_3$ (Walker et al., 2011; Clarisse et al., 2013). More details on their construction will be provided in Section 3.2.



In the present study, we are interested in characterizing the instrumental performance for retrieving $NH_3$ column abundances. $\sigma_{noise}$ and $\sigma_{abs}$ constitute two suitable metrics. The former quantifies the propagation of the instrumental noise to the retrieval and is a firm lower bound to the total measurement uncertainty. The quantity is easy to calculate as it relies only on a Jacobian and an estimate of the noise covariance matrix. $\sigma_{abs}$, on the other hand, includes uncertainty related to spectral interferences, irrespective of whether they are explicitly taken into account through fitting in the retrieval procedure. The capability of differentiating such interferences from the $NH_3$ spectral signature is directly related to the instrument. For example, at higher spectral resolution, slowly varying surface emissivity is expected to be more easily disentangled from the sharper $NH_3$ spectral features. As such, this metric is fully representative of how the instrumental performances propagate to the retrieval uncertainty. It is the main metric used in this study.

There are two important notes to be made, both related to the fact that in the above we assumed that $\boldsymbol{K}$ is known. $\boldsymbol{K}$ inherently depends on the state of the surface and the atmosphere, and more particularly on the thermal contrast (TC), which is the temperature difference between the surface and the ambient temperature at the location where the bulk of the $NH_3$ is located. TC is crucial in the infrared as it determines whether the spectral signature of the species is observed in absorption or emission, and the strength of this signal. We refer to Di Gioacchino et al. (2024); Noppen et al. (2023) for a detailed discussion on TC.

The uncertainty on the scene results in an additional contribution $\sigma_{rel}$ to the total retrieval uncertainty $\sigma_{total}$, with (Clarisse et al., 2023)

$$\sigma_{total}^2 = \sigma_{abs}^2 + \sigma_{rel}^2. \tag{7}$$

The notation *abs* and *rel* refer to the fact that $\sigma_{abs}$ is an absolute contribution to the uncertainty, independent of the column, while $\sigma_{rel}$ is a relative contribution that varies linearly with the column $C_{NH_3}$. It was shown in Clarisse et al. (2023) that $\sigma_{rel}$ is also, to a first order, inversely proportional to the TC:

$$\sigma_{rel} \approx \left( 0.07 + \frac{1.6\,\text{K}}{\text{TC}} \right) C_{NH_3}. \tag{8}$$

This formula was derived statistically assuming a Gaussian-distributed $NH_3$ vertical profile peaking at the surface with width $h$, and the TC defined as the difference between the surface temperature and that at an altitude of $h/2$. Uncertainties of the order of 1–2 K on both surface and air temperature were assumed. In what follows, we will assume that the input parameters used to calculate $\boldsymbol{K}$, in particular the temperatures, are provided from an external source, like the output of a meteorological model, and are not retrieved from the measurement itself. Uncertainty on the TC is, in this case, not related to the instrument, so this source of uncertainty can be disregarded when comparing different hypothetical instruments. The design of an instrument that is simultaneously capable of retrieving temperatures and $NH_3$ (like IASI) will not be studied here. That being said, for the instruments under investigation, surface temperature can always be estimated from measurements as we consider sounders measuring in the atmospheric window between 900 and 1000 cm$^{-1}$.

The second note to be made is that both $\sigma_{noise}$ and $\sigma_{abs}$ depend on the atmospheric state (and again mostly the TC) because $\boldsymbol{K}$ does. It is therefore important to specify the exact atmosphere under which these numbers are calculated. In the rest of





the paper, $\boldsymbol{K}$ is calculated for a standard atmosphere (Kneizys et al., 1996), a Gaussian-distributed $NH_3$ vertical profile with $h = 1.35$ km, and a surface temperature such that the TC is 10 K. For IASI observations, it was shown that (Clarisse et al., 2023)

$$\sigma_{\mathrm{abs}}(\mathrm{TC}) \approx \frac{3.6 \times 10^{16}}{\mathrm{TC}} \frac{\mathrm{molec.K}}{\mathrm{cm}^2}. \tag{9}$$

For our IASI reference scene (see Figure 2 discussed in Section 4.2), we obtained $\sigma_{\mathrm{abs}} \approx 3.6 \times 10^{15}$ molec.cm$^{-2}$, which is consistent with this value for a TC of 10 K. These fixed scenario numbers allow for a direct comparison of instruments, as the value of the TC depends on the state of the scene, not the instrument. However, if desired, the retrieval uncertainty at other TC can be obtained by scaling $\sigma_{\mathrm{abs}}$ inversely proportional to the TC.

### 3.2 The matched filter (MF)

A quantity related to $\sigma_{\mathrm{abs}}$ is the signal-to-noise or signal-to-clutter ratio

$$\frac{\hat{x}}{\sigma_{\mathrm{abs}}} = \mathrm{MF} = \frac{\boldsymbol{K}^T \mathbf{S}_g^{-1} (\boldsymbol{y} - \boldsymbol{y}_g)}{\sqrt{\boldsymbol{K}^T \mathbf{S}_g^{-1} \boldsymbol{K}}}. \tag{10}$$

This quantity, known as the (adaptive) matched filter, is widely adopted in scientific domains involving signal processing. This includes hyperspectral imaging (Manolakis et al., 2016) and satellite remote sensing (Walker et al., 2011; Clarisse et al., 2013), where it is sometimes referred to as the hyperspectral range index (HRI, Van Damme et al. (2014)). By definition, the distribution of the MFs for the background spectra (the clutter) has a mean of 0 and a variance of 1. By imposing a threshold on its value, the MF can be used as a detection tool to identify the presence of the target signature in an observed spectrum. Even if the set of spectra from which the pair $(\mathbf{S}_g, \boldsymbol{y}_g)$ is estimated does not follow a multivariate normal distribution, the distribution of the MF values on the background spectra is in practice often near normal (Schaum, 2021). This implies that an MF value above 2.5 in absolute value indicates that the observed spectrum contains a signature of the target species with a probability larger than 98.5%. Conversely, it means that 98.5% of the spectra that were used for the construction of $\mathbf{S}_g$ and $\boldsymbol{y}_g$ have an MF value (in absolute value) below 2.5.

Key to the calculation of $\sigma_{\mathrm{abs}}$ and MF is the construction of $\mathbf{S}_g$ and $\boldsymbol{y}_g$. As mentioned above, they are most frequently estimated from a large collection of observed spectra. Ideally, these should contain no detectable signature of the target species, as this leads to a decrease in performance (Manolakis et al., 2016). Clarisse et al. (2013); Kim et al. (2016) proposed an iterative approach where the detection itself is used to exclude spectra with detectable signatures of the target species. A first MF value is calculated from the pair $(\mathbf{S}_g, \boldsymbol{y}_g)$ obtained from all observed spectra in the spectral range of interest. Next, a new pair $(\mathbf{S}_g, \boldsymbol{y}_g)$ is estimated, but this time only for spectra with a low $|\mathrm{MF}|$ value. This process is repeated until convergence. As explained in Franco et al. (2018), using a very low threshold increases the sensitivity of the $NH_3$ detection but leads to larger values of MF over the background too. Therefore, MF is explicitly renormalized at each iteration to have a standard deviation of exactly 1 on a well-chosen area of the scene with low MF values. Finally, to prevent numerical problems associated with the very small eigenvalues of $\mathbf{S}_g$, $\mathbf{S}_g^{-1}$ is calculated using the methodology presented in Clarisse et al. (2023).



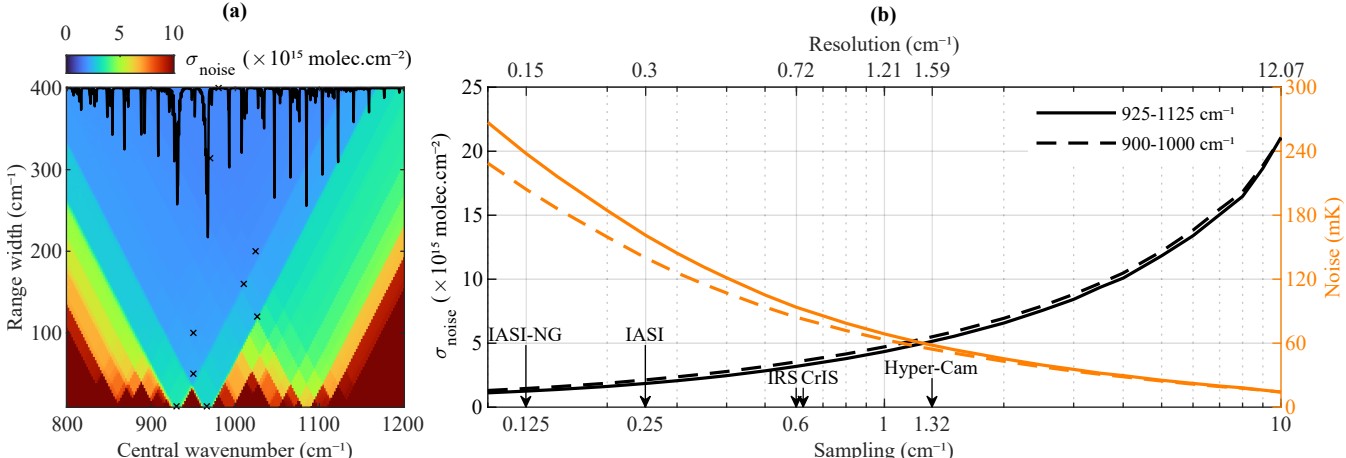

**Figure 1.** (a) Measurement uncertainty ($\sigma_{\mathrm{noise}}$) for an instrumental noise ($\epsilon$) of 100 mK from data simulated at a spectral resolution of 0.3 cm$^{-1}$ as a function of central wavenumber and range width. The NH$_3$ Jacobian $\boldsymbol{K}$ for the full range is indicated on top (black line). The crosses indicate the ranges discussed in Table 1. (b) On the left axis, $\sigma_{\mathrm{noise}}$ is drawn as a function of the spectral sampling/resolution for two different spectral ranges. On the right axis, $\epsilon$ is shown corresponding to a $\sigma_{\mathrm{noise}} = 3 \times 10^{15}$ molec.cm$^{-2}$. The spectral samplings of some existing and planned spaceborne instruments are indicated, along with the sampling of the Hyper-Cam airborne measurements (Crevoisier et al., 2014; Glumb et al., 2003; Holmlund et al., 2021).

### 3.3 Signal-to-noise (SNR) and false alarm rate (FAR)

The quantities $\sigma_{\mathrm{abs}}$ and MF convey similar information for comparing instrumental performance when the pair ($\mathbf{S}_g$, $\boldsymbol{y}_g$), on which they both rely, represents a multivariate normal background distribution. Consistent with its interpretation as a signal-to-noise ratio, an MF value of 1 corresponds to an NH$_3$ value of $x = \sigma_{\mathrm{abs}}$ (this can be seen by substituting $\boldsymbol{y} - \boldsymbol{y}_g = x\boldsymbol{K}$ in Equation 10). Larger values of MF can be interpreted likewise in units of $\sigma_{\mathrm{abs}}$. However, while $\sigma_{\mathrm{abs}}$ is a single number calculated from an ensemble of spectra, MF can be calculated on individual spectra and can be used to investigate possible departures

from linearity or non-Gaussian statistics, which might be instrument-dependent. For this reason, we use two additional metrics based on MF in this study.

The first is the mean signal-to-noise ratio (SNR), calculated as the average MF value of a given set of spectra containing the target signature. In view of the above, the expected behavior of this SNR is inversely proportional to $\sigma_{\mathrm{abs}}$.

Another common metric based on the MF is the false alarm rate (FAR, Manolakis et al. (2016)). Rather than assessing the

performance on spectra containing observable quantities of NH$_3$, this metric evaluates how likely it is that noise or spectral interferences are misinterpreted as NH$_3$. It is defined as the fraction of false detections over the total number of observations for a given set of NH$_3$-free spectra.





**Table 1.** $\sigma_{\text{noise}}$ for the spectral ranges marked with crosses in Figure 1a and, in the last column, $\sigma_{\text{noise}}$ relative to the range used for the IASI NH$_3$ retrieval. An instrumental noise of 100 mK is assumed.

| Range (width) in cm$^{-1}$ | $\sigma_{\text{noise}}$ (molec.cm$^{-2}$) | Relative to IASI (%) |
|---|---|---|
| 780–1180 (400) | $1.72 \times 10^{15}$ | 99.5 |
| 812–1126 (314) | $1.73 \times 10^{15}$ | 100 |
| 925–1125 (200) | $1.85 \times 10^{15}$ | 107 |
| 930–1090 (160) | $2.04 \times 10^{15}$ | 118 |
| 965–1085 (120) | $2.61 \times 10^{15}$ | 151 |
| 900–1000 (100) | $2.13 \times 10^{15}$ | 123 |
| 925–975 (50) | $2.27 \times 10^{15}$ | 131 |
| 961–971 (10) | $3.06 \times 10^{15}$ | 177 |
| 925–935 (10) | $3.45 \times 10^{15}$ | 199 |

## 4 Results

In the first part of this section, we analyze the performance of hypothetical instruments with different spectral ranges and
resolutions, focusing on the $\sigma_{\text{noise}}$ metric. Then we move to the other metrics (SNR, FAR and $\sigma_{\text{abs}}$), for which we rely on manipulations of real data from IASI (Section 4.2) and Hyper-Cam (Section 4.3).

### 4.1 From simulated spectra

In this section, Fourier Transform-type instruments were assumed, with different maximum optical path differences (MOPD) and spectral ranges. The MOPD determines both the spectral sampling (0.5/MOPD) and spectral resolution (0.60335/MOPD)
(Davis et al., 2005). The instrumental noise was assumed to be constant across the spectrum with $\epsilon$ in radiance units corresponding to 100 mK for a reference blackbody at 288 K and 1000 cm$^{-1}$. In this case, $\mathbf{S}_\epsilon$ is proportional to the identity matrix as $\mathbf{S}_\epsilon = \epsilon^2 \mathbf{I}$ and

$$\sigma_{\text{noise}} = \epsilon \left( \boldsymbol{K}^T \boldsymbol{K} \right)^{-1/2} . \tag{11}$$

The line-by-line radiative transfer code Atmosphit was used as the forward model (Barret et al., 2005; Coheur et al., 2005)
under the fixed standard atmospheric conditions specified in Section 3.1. We focus on the 800–1200 cm$^{-1}$ range that contains most of the $\nu_2$ band of NH$_3$, showing strong features around 931 and 967 cm$^{-1}$, as can be seen from the Jacobian $\boldsymbol{K}$ displayed in Figure 1a.

We first evaluate the effect of spectral range on the measurement uncertainty, for an instrument with an MOPD of 2 (sampled at 0.25 cm$^{-1}$ and with a spectral resolution of 0.3 cm$^{-1}$). Figure 1a shows $\sigma_{\text{noise}}$ as a function of both central wavenumber and
range width. Table 1 lists a series of selected spectral ranges of various widths (marked with crosses in Figure 1a) along with



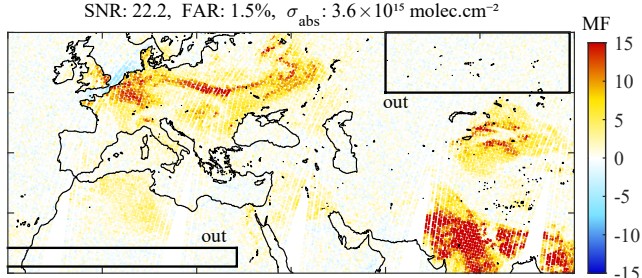

**Figure 2.** MF$_{\text{IASI}}$ distribution calculated from the morning IASI overpass on April 8, 2020 from Clarisse et al. (2023). The signal-to-noise ratio (SNR), the false alarm rate (FAR) and the uncertainty ($\sigma_{\text{abs}}$) are indicated on top. The "out" boxes correspond to the areas used to estimate the FAR.

their $\sigma_{\text{noise}}$. The last column of Table 1 expresses retrieval uncertainty relative to the one for the range used for the NH$_3$-IASI retrieval (812–1126 cm$^{-1}$).

As expected, we observe the smallest uncertainties for the largest range, with the 925–975 cm$^{-1}$ interval contributing the most. Compared to the IASI retrieval range, we find that extending the range to 400 cm$^{-1}$ improves the uncertainty by only
0.5%. For range widths of 200 cm$^{-1}$ and 100 cm$^{-1}$, $\sigma_{\text{noise}}$ increases by 7% and 23%, respectively. Increases greater than 60% are observed for ranges shorter than 35 cm$^{-1}$, as these contain only one of the two major features of the NH$_3$ signature. The range focusing on the feature centered at 967 cm$^{-1}$ is more sensitive to NH$_3$ than the second one centered on 931 cm$^{-1}$, as seen from the last two rows of Table 1.

The effect of spectral sampling is investigated for two ranges. The first is 200 cm$^{-1}$ wide, extending from 925 to 1125 cm$^{-1}$,
and the second is 100 cm$^{-1}$ wide between 900 and 1000 cm$^{-1}$. Figure 1b shows $\sigma_{\text{noise}}$ (black curves) calculated for both ranges as a function of the spectral sampling, which varies from 0.1 to 10 cm$^{-1}$ (the equivalent spectral resolution is indicated on top). For the 900–1000 cm$^{-1}$ range, degrading the sampling from 0.1 to 1 cm$^{-1}$ increases $\sigma_{\text{noise}}$ by a factor of 3.6. When the sampling is further degraded from 1 to 10 cm$^{-1}$, $\sigma_{\text{noise}}$ increases by a further factor of 4.5. Consistent with what was observed above, $\sigma_{\text{noise}}$ is slightly larger for the shortest range. However, this difference is small, and the effect of the spectral
sampling/resolution is clearly greater than that of the spectral range.

The right axis of the graph shows the required noise $\epsilon$ (in orange) to obtain a fixed value of $\sigma_{\text{noise}} = 3 \times 10^{15}$ molec.cm$^{-2}$. This was calculated by inversion of Equation 11. As can be seen from that relation, in principle, any desired retrieval uncertainty can be achieved as long as the instrumental noise is small enough. However, at very coarse resolution, the NH$_3$ signature becomes difficult to distinguish from other spectral contributions. Hence the need to look at simulations involving real data and
the $\sigma_{\text{abs}}$ metric.





**Figure 3.** SNR, FAR and $\sigma_{abs}$ as a function of the number of channels for the degraded IASI (a) and the Hyper-Cam data over Rosignano (b) and around the Po Valley (c). The triangles show the results for continuous channels between 900 and 1000 $\mathrm{cm^{-1}}$, while the dots represent for the well-chosen spectral bands of different widths, as indicated by their respective color for IASI / Hyper-Cam. The open triangles on $\sigma_{abs}$ plots indicate $\sigma_{noise}$.

## 4.2 From IASI spaceborne data

### 4.2.1 Uniformly spaced channels

Taking advantage of the high spectral resolution of the native IASI measurements, we degraded the original L1C spectra to characterize hypothetical sounders with lower resolution. The sampling was degraded to 1, 2, 5 and 10 $\mathrm{cm^{-1}}$ by block-averaging neighboring channels of the original spectra. In view of the results of the previous section, the spectral range was





**Figure 4.** Each panel shows an MF distribution calculated from degraded IASI spectra. They show distributions retrieved for an instrument with a sampling of 1 cm$^{-1}$ (left column) and 2 cm$^{-1}$ (right column). The first row is for an instrument with continuous channels between 900 and 1000 cm$^{-1}$, the other rows for spectra with (b) 5, (c) 4, (d) 3, (e) 2 spectral bands, respectively. The SNR and the FAR (in %) are noted on the upper right of each panel along with $\sigma_{\mathrm{abs}}$ (in molec.cm$^{-2}$).





**Figure 5.** Each panel shows an MF distribution calculated from degraded IASI spectra. They show distributions retrieved for an instrument with a sampling of 5 cm$^{-1}$ (left column) and 10 cm$^{-1}$ (right column). The first row is for an instrument with continuous channels between 900 and 1000 cm$^{-1}$, the other rows for spectra with (b) 5, (c) 4, (d) 3, (e) 2 spectral bands, respectively. The SNR and the FAR (in %) are noted on the upper right of each panel along with $\sigma_{abs}$ (in molec.cm$^{-2}$).





**Table 2.** SNR and FAR (%), along with $\sigma_{\mathrm{abs}}$ ($\times 10^{16}$ molec.cm$^{-2}$) and $\sigma_{\mathrm{noise}}$ ($\times 10^{16}$ molec.cm$^{-2}$) for downgraded IASI spectra with $n$ continuous channels between 900 and 1000 cm$^{-1}$, and with 5, 4, 3, 2 bands of different width (see Table 3).

|  |  | 1 cm$^{-1}$ | 2 cm$^{-1}$ | 5 cm$^{-1}$ | 10 cm$^{-1}$ |
|---|---|---|---|---|---|
| **$n$ bands** | SNR | 16.84 | 11.43 | 5.76 | 2.39 |
|  | FAR | 2.25 | 1.71 | 1.49 | 2.81 |
|  | $\sigma_{\mathrm{abs}}$ | 0.63 | 0.98 | 1.99 | 4.06 |
|  | $\sigma_{\mathrm{noise}}$ | 0.59 | 0.86 | 1.59 | 2.66 |
| **5 bands** | SNR | 10.37 | 8.12 | 5.46 | 3.07 |
|  | FAR | 1.34 | 1.61 | 1.48 | 1.31 |
|  | $\sigma_{\mathrm{abs}}$ | 1.10 | 1.37 | 2.01 | 3.75 |
|  | $\sigma_{\mathrm{noise}}$ | 0.92 | 1.07 | 1.52 | 2.43 |
| **4 bands** | SNR | 9.27 | 7.29 | 4.70 | 2.76 |
|  | FAR | 1.43 | 1.39 | 1.82 | 1.39 |
|  | $\sigma_{\mathrm{abs}}$ | 1.21 | 1.53 | 2.24 | 3.90 |
|  | $\sigma_{\mathrm{noise}}$ | 0.92 | 1.07 | 1.52 | 2.57 |
| **3 bands** | SNR | 7.95 | 6.33 | 4.55 | 2.43 |
|  | FAR | 1.26 | 1.44 | 1.32 | 2.03 |
|  | $\sigma_{\mathrm{abs}}$ | 1.36 | 1.72 | 2.54 | 4.52 |
|  | $\sigma_{\mathrm{noise}}$ | 1.17 | 1.38 | 1.95 | 3.55 |
| **2 bands** | SNR | 5.74 | 5.25 | 2.69 | 1.50 |
|  | FAR | 1.64 | 1.64 | 1.99 | 2.56 |
|  | $\sigma_{\mathrm{abs}}$ | 2.04 | 2.08 | 4.00 | 5.99 |
|  | $\sigma_{\mathrm{noise}}$ | 1.46 | 1.38 | 2.56 | 3.70 |

limited to the 900–1000 cm$^{-1}$ region. Averaging multiple channels together reduces the single-channel noise, and to be able to fairly compare the different hypothetical instruments, artificial noise was added to each spectrum to match the noise level of the spectra degraded to 1 cm$^{-1}$ sampling. The details of how the spectra were degraded, and how the noise levels were estimated are explained in the Appendix A. There, we also show that the spectra obtained in this way have a spectral resolution that matches very closely their spectral sampling, so that both terms will be used interchangeably in this section.

As the basis for our analysis, we used all IASI morning observations of a single day (8 April 2020) between -65° and 65° latitude. The original MF$_{\mathrm{IASI}}$ from Clarisse et al. (2023) is illustrated in Figure 2 over an area covering part of Europe, northern Africa and western Asia. For clarity, we only show this region in the following. However, since we use all data between -65°





and 65° latitude, the results that will be presented are based on and representative of global data (except for the poles). The pair

$(\mathbf{S}_g, \boldsymbol{y}_g)$, the MF and the four performance metrics were determined as follows:

**$(\mathbf{S}_g, \boldsymbol{y}_g)$** These were constructed from all degraded spectra with a corresponding $|\mathrm{MF}_{\mathrm{IASI}}| \leq 1.5$.

**MF** MF was calculated from the degraded spectra with Equation 10, using $(\mathbf{S}_g, \boldsymbol{y}_g)$ defined above and $\boldsymbol{K}$ calculated from forward simulations as explained in Section 3, and block-averaged in a similar way as the spectra. For the reason explained in Section 3.2, the MF was renormalized to have a standard deviation of 1 over the area east of -140°W (remote Pacific

Ocean).

**SNR** A mean SNR was calculated from the average MF for spectra with $\mathrm{MF}_{\mathrm{IASI}} > 20$. The SNR for $\mathrm{MF}_{\mathrm{IASI}}$ on this set is 22.2.

**FAR** The FAR was calculated as the fraction of observations for which $|\mathrm{MF}| > 2.5$ for observations inside the "out" boxes defined on Figure 2. As mentioned above, when the MF follows a normal distribution, the expected value of the FAR is 1.5% for this threshold (this is the case for $\mathrm{MF}_{\mathrm{IASI}}$).

**$\sigma_{\mathbf{abs}}$** $\sigma_{\mathrm{abs}}$ was straightforwardly calculated from Equation 6 and estimated to $3.6 \times 10^{15}$ molec.cm$^{-2}$ for $\mathrm{MF}_{\mathrm{IASI}}$.

**$\sigma_{\mathbf{noise}}$** $\sigma_{\mathrm{noise}}$ was calculated from Equation 3. While the single channel noise is defined to be the same across all degradations, $\sigma_{\mathrm{noise}}$ varies because of the different Jacobians and number of channels.

The metrics are summarized in column (a) of Figure 3 (triangles) and Table 2, while the MF distributions are shown in rows (a) of Figures 4 and 5. Note that the color scale of each panel was adapted to show visually similar $NH_3$ over India, enabling

better comparison of the noise levels.

The distributions from the spectra sampled at 1, 2 and 5 cm$^{-1}$ visually match the original distribution closely (see Figure 2), demonstrating that high spectral resolution is not a requirement for reliable $NH_3$ measurements. The main difference is in the level of observed noise, as witnessed by the progressively higher $\sigma_{\mathrm{abs}}$ and lower SNR numbers for coarser samplings. From 100 channels of 1 cm$^{-1}$ to 20 channels of 5 cm$^{-1}$, the noise increases by a factor of approximately 3.

The FARs are all relatively close to the theoretical value of 1.5%, a consequence of the MF normalization and the Gaussian nature of the MF distribution. The configuration of 10 channels of 10 cm$^{-1}$ sampling (top-right panel of Figure 5) has the highest FAR (2.8%). While the main $NH_3$ patterns of the original distribution are still visible with an SNR of 2.4, slightly negative values are seen over North Africa and the Middle East. These can be attributed to variations in surface emissivity, which apparently cannot be resolved with the given spectral resolution and which also explain the higher FAR.

As for $\sigma_{\mathrm{noise}}$ (open triangles in Figure 3), the results are consistent with the theoretical simulations from Section 4.1, increasing by a factor of 4.5 between 1 and 10 cm$^{-1}$ sampling. In all cases, $\sigma_{\mathrm{abs}}$ is larger than $\sigma_{\mathrm{noise}}$ as expected. We also see that at 1 cm$^{-1}$ resolution, there is hardly any difference between the $\sigma_{\mathrm{abs}}$ and $\sigma_{\mathrm{noise}}$. However, as the resolution gets coarser, the difference becomes progressively larger. This is a consequence of the fact that at coarser resolution it becomes increasingly difficult to unambiguously distinguish $NH_3$ from other interfering physical parameters. At 10 cm$^{-1}$, the contribution of these

spectral interferences is $\sqrt{\sigma_{\mathrm{abs}}^2 - \sigma_{\mathrm{noise}}^2} \approx 3 \times 10^{16}$ molec.cm$^{-2}$, limiting inherently the unambiguous identification of columns of this magnitude, even for an instrument with a very small radiometric noise.



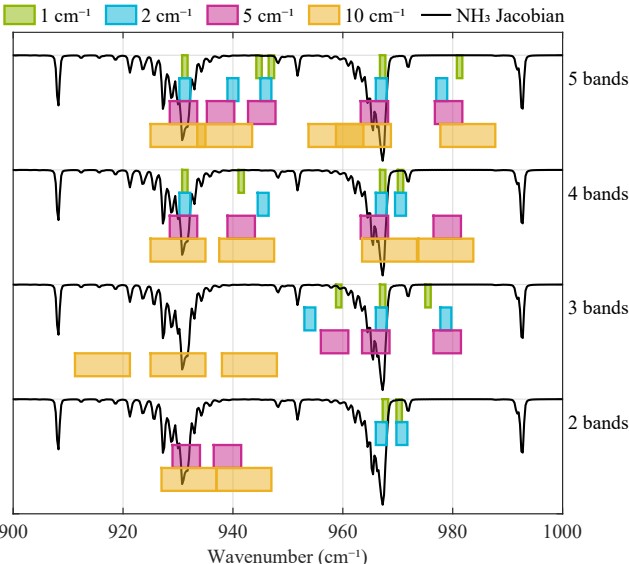

**Figure 6.** Illustration of the optimal spectral bands of various widths (from 1 to 10 cm$^{-1}$) for observing atmospheric NH$_3$ in the infrared. The NH$_3$ Jacobian from Figure 1 is indicated as well in each subpanel (black line). Exact band positions are listed in Table 3.

### 4.2.2 Well-chosen bands

Especially for the instruments with 5 and 10 cm$^{-1}$ samplings, there are potentially better choices for the channel centers in relation to the absorption features of NH$_3$. In this section, we explore again potential instruments with channels 1, 2, 5, and
10 cm$^{-1}$ wide. However, instead of instruments with uniform sampling, we investigate instruments with fewer bands (2 to 5) and optimize their positions within the 900–1000 cm$^{-1}$ interval. This could also potentially further reduce the complexity of a dedicated NH$_3$ sounder. The locations of the bands reported here were determined as those that minimized the resulting $\sigma_{\mathrm{abs}}$ (calculated using the same procedure as before). For 2 bands, an exhaustive search was performed, while for 3, 4, and 5 bands, these were found with a stochastic hill climbing algorithm (Russell and Norvig, 2010) starting from 300 randomly
chosen configurations. The optimal bands are summarized in Table 3 and illustrated in Figure 6.

As shown in the figure, for configurations with 2 or 3 bands, bands are always selected very close to each other, with one band targeting either the 931 cm$^{-1}$ or 967 cm$^{-1}$ NH$_3$ feature, and one or two bands focusing on the baseline. When more bands are available, the two strong NH$_3$ features are always targeted. For the configurations with more bands, bands are relatively close together as well and the full 900–1000 cm$^{-1}$ range is never exploited. This likely helps avoiding broadband variations in the
baseline due to surface emissivity or clouds.

The associated MF distributions are shown in Figures 4 and 5, and the statistics are summarized in column (a) of Figure 3 (dots) and Table 2. All distributions resemble the original distribution, all reproducing the main NH$_3$ enhancements over Europe, Russia, North Africa, India and Myanmar. However, with an SNR of 1.5, the 2-band 10 cm$^{-1}$ configuration is by far the noisiest, and with a measurement uncertainty of $6\times10^{16}$ molec.cm$^{-2}$ such an instrument is only suitable for measuring





**Table 3.** Table summarizing the position of the optimal bands for observing atmospheric $NH_3$ for the test data under consideration. Each value represents the center of the band in $\mathrm{cm}^{-1}$.

|  | $1\,\mathrm{cm}^{-1}$ | $2\,\mathrm{cm}^{-1}$ | $5\,\mathrm{cm}^{-1}$ | $10\,\mathrm{cm}^{-1}$ |
|---|---|---|---|---|
| **5 bands** | 931.25 | 931.25 | 931.00 | 930.00 |
|  | 944.75 | 940.00 | 937.75 | 938.50 |
|  | 947.00 | 946.00 | 945.25 | 958.75 |
|  | 967.25 | 967.00 | 965.75 | 963.75 |
|  | 981.25 | 978.00 | 979.25 | 982.75 |
| **4 bands** | 931.25 | 931.25 | 931.00 | 930.00 |
|  | 941.50 | 945.50 | 941.50 | 942.50 |
|  | 967.25 | 967.00 | 965.75 | 968.50 |
|  | 970.50 | 970.50 | 979.00 | 978.75 |
| **3 bands** | 959.25 | 954.00 | 958.50 | 916.25 |
|  | 967.25 | 967.00 | 966.00 | 930.00 |
|  | 975.50 | 978.75 | 979.00 | 943.00 |
| **2 bands** | 967.75 | 967.00 | 931.50 | 932.00 |
|  | 970.25 | 970.75 | 939.00 | 942.00 |

very large $NH_3$ columns. As expected, there is a clear general decrease in SNR and an increase in $\sigma_{\mathrm{abs}}$ for wider bands, with a factor of 3–4 between 1 and $10\,\mathrm{cm}^{-1}$ for a given number of bands. The number of bands also affects these two metrics, but to a lesser extent —about a factor of 2 between 2 and 5 bands. As a general observation, it is more efficient to observe $NH_3$ using fewer bands at finer sampling than more bands at coarser sampling. For instance, in terms of SNR and $\sigma_{\mathrm{abs}}$, we observe that even just 2 bands of $1\,\mathrm{cm}^{-1}$ perform similarly to 5 bands of $5\,\mathrm{cm}^{-1}$. Likewise, 2 bands of $2\,\mathrm{cm}^{-1}$ outperform 5 bands

of $10\,\mathrm{cm}^{-1}$. We also see from the statistics in Table 2 that for the coarsest spectral resolutions of 5–$10\,\mathrm{cm}^{-1}$, the precise location of the bands is key, and that the optimal choice provides a genuine gain in performance. For instance, 5 well-chosen bands of $10\,\mathrm{cm}^{-1}$ outperform 10 contiguous bands between 900–$1000\,\mathrm{cm}^{-1}$. For the $10\,\mathrm{cm}^{-1}$ band configuration, even just 3 well-chosen bands match the performance of the latter.

     The MF does a good job of minimizing the FAR, with most configurations having a FAR below 2%. It is notable that the 5

well-chosen bands of $10\,\mathrm{cm}^{-1}$ have a much better FAR and performance over deserts than the 10 uniformly sampled bands of $10\,\mathrm{cm}^{-1}$. This demonstrates that the optimized choice of band centers not only helps increase the SNR but can also help mitigate false detections. In terms of FAR, besides the 10 bands of $10\,\mathrm{cm}^{-1}$, the configurations that perform the poorest are the 2 or 3 bands of $10\,\mathrm{cm}^{-1}$, with a FAR between 2% and 2.6% (see rows (d) and (e) of Figures 4 and 5). These are the




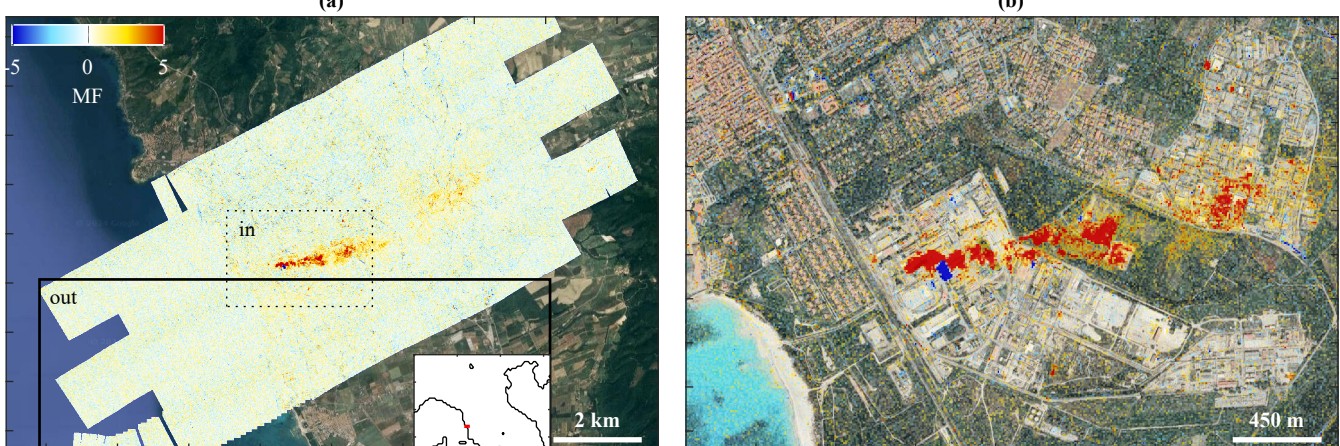

**Figure 7.** (a) Superimposed on satellite visible imagery from © Google Maps, MF distribution retrieved from Hyper-Cam measurements performed over Rosignano (the location within Italy is shown in red in the inset) on 19 May 2022, revealing a plume of $NH_3$ originating from the soda ash plant. The "in" and "out" boxes correspond to the areas used to estimate the SNR and FAR, and to renormalize MF. (b) Close-up view of the region outlined by the dotted rectangle in the left panel.

same configurations that exhibit the largest negative values over the deserts in North Africa and the Middle East. For these, the information content of the measurement is not large enough to distinguish surface effects from the $NH_3$ spectral signature. Related to this, we again observe that the difference between $\sigma_{abs}$ and $\sigma_{noise}$ increases as the spectral resolution decreases (see Table 2). Here, we additionally see that these differences increase as the number of bands decreases, showing that both the number of bands and their resolution affect the capability of an instrument to disentangle the $NH_3$ signature from other spectral interferences.

### 4.3 From Hyper-Cam airborne measurements

#### 4.3.1 Uniformly spaced channels

As explained in Section 2, the Hyper-Cam data were spatially averaged to reduce noise. Even after averaging, the resulting resolution of $100\,\mathrm{m}^2$ is still 1000 times finer than IASI's ($> 100\,\mathrm{km}^2$). At such a fine resolution, the measurements are expected to exhibit much stronger variations in surface emissivity due to features like rooftops, solar panels, crop fields, or bare soil. Such features are not seen individually by IASI and might affect $NH_3$ measurements at low spectral resolution. Hence, in this section, we explore the effects of spectrally degrading the Hyper-Cam measurement data to verify whether the results from the previous section remain valid at smaller spatial scales.

From the two surveys (over Rosignano and around the Po Valley, see Section 2), MF distributions were calculated from the observed spectra covering the spectral range from 870 to $1260\,\mathrm{cm}^{-1}$ at the native $1.3\,\mathrm{cm}^{-1}$ sampling. In both cases, the pair $(\mathbf{S}_g, \boldsymbol{y}_g)$ was built in 8 iterations, each time filtering out pixels with $|\mathrm{MF}| > 1.5$. The MF distributions were renormalized using





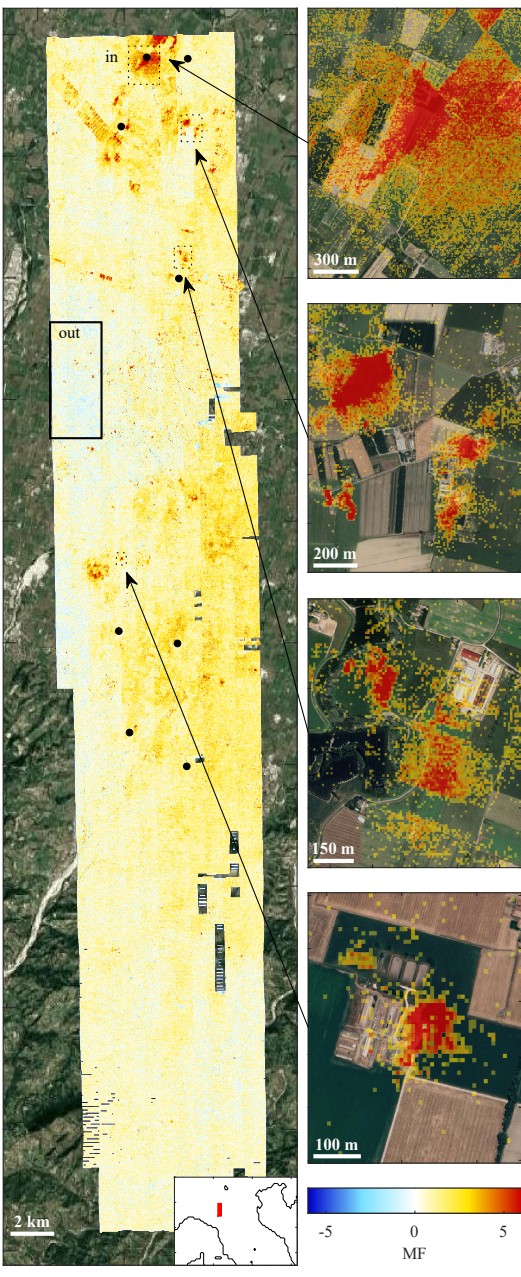

**Figure 8.** Superimposed on satellite visible imagery from © Google Maps, MF distribution retrieved from Hyper-Cam measurements performed over the Po valley (the location within Italy is shown in red in the inset) on 30 June 2022, revealing $NH_3$ hotspots over various farms. Some of these identified sources, delimited by dotted rectangles, are displayed on the right side of the figure. The black dots refer to the agricultural sources reported in the E-PRTR between 2020 and 2022. The "in" and "out" boxes correspond to the areas used to estimate the SNR and FAR, and to renormalize MF.



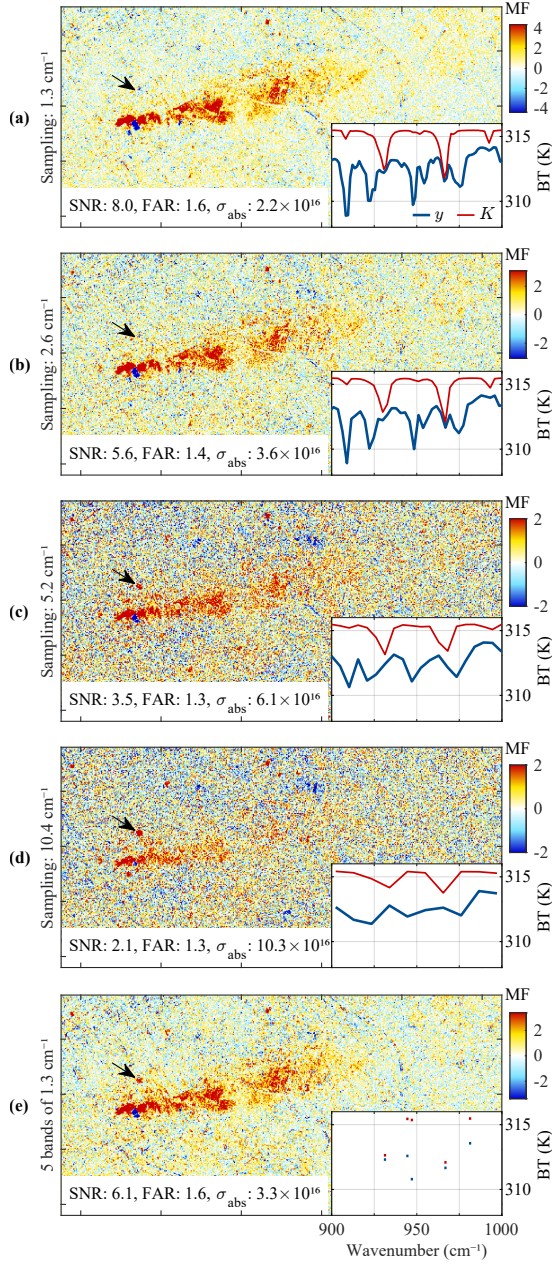

**Figure 9.** MF distributions calculated from Hyper-Cam measurements in Rosignano downgraded to a spectral sampling of (a) 1.3, (b) 2.6, (c) 5.2 and (d) 10.4 $\mathrm{cm}^{-1}$ between 900 and 1000 $\mathrm{cm}^{-1}$. Panel (e) corresponds to the distribution obtained from spectra computed in five well-chosen bands of 1.3 $\mathrm{cm}^{-1}$ width. The arrows indicate a false signal of $NH_3$. The SNR, FAR (in %) and $\sigma_{\mathrm{abs}}$ (in molec.$\mathrm{cm}^{-2}$) are indicated on the bottom-left of each panel. The bottom-right shows the average of the 500 downsampled spectra observed over the plume used to calculate the SNR (in blue) and the $NH_3$ Jacobian (in red).





**Figure 10.** Panel (a) shows the MF distribution calculated from Hyper-Cam measurements over the Po Valley downgraded to a spectral sampling of $1.3\ \mathrm{cm^{-1}}$ between 900 and $1000\ \mathrm{cm^{-1}}$, superimposed on satellite visible imagery from © Google Maps. Panels (b), (c), and (d) represent the equivalent MF distributions calculated from spectra degraded as 5, 4, and 3 well-defined bands of $1.3\ \mathrm{cm^{-1}}$. The SNR, FAR (in %) and $\sigma_{\mathrm{abs}}$ (in $\mathrm{molec.cm^{-2}}$) are indicated at the bottom right of each panel.



the spectra measured over the areas referred to with "out" in Figures 7 and 8, defining the background. In the measurements over Rosignano, we observe an $NH_3$ plume originating from the plant, extending over more than 2 km in the eastward direction; see Figure 7. As can be seen from the zoomed-in view of the industrial complex shown in panel (b) of the figure, both positive and negative MF values are observed. The sign is related to the sign of the TC, which determines whether the spectral signature

is observed in absorption or emission (Noppen et al., 2023). The MF distribution for the scene in and around the Po Valley is shown on the left side of Figure 8. Various hotspots of $NH_3$ can be observed over farms, some of which are shown in close-ups on the right side of the figure. The agricultural sources listed in the E-PRTR between 2020 and 2022 are indicated by black dots, most of which are visible in the MF distribution.

For the evaluation of the results, we followed as much as possible the approach applied to the IASI spectra. The spectral

range was as before limited to 900–1000 $cm^{-1}$ and spectra were degraded by block-averaging neighboring channels of the original measurements. Given the initial sampling of 1.3 $cm^{-1}$, this procedure results in spectra sampled at 2.6, 5.2, and 10.4 $cm^{-1}$, which is close to the 2, 5 and 10 $cm^{-1}$ studied earlier. For the same reasons as above, Gaussian noise was added to each spectrum to match the native unapodized noise of the 1.3 $cm^{-1}$ spectra, estimated at $4.43 \times 10^{-4}$ W.(m$^2$.sr.cm$^{-1})^{-1}$ (see Appendix A). For each configuration, ($\mathbf{S}_g$, $\boldsymbol{y}_g$) were calculated based on the pixels for which $|MF| < 1.5$ of the original

spectra. The SNR was calculated from the 500 pixels presenting the highest MFs located in the "in" boxes defined in Figures 7 and 8. The FAR was calculated as the fraction of spectra with $|MF| > 2.5$ within the "out" boxes defined in Figures 7 and 8.

Panels (a) to (d) of Figure 9 show a close-up of the resulting MF distributions over Rosignano. As before, the color scale was adapted so that the $NH_3$ plume appears similar in each panel. The SNR, FAR, and $\sigma_{abs}$ are indicated on the bottom-left of each panel and are also shown in column (b) of Figure 3 (triangles). As expected, the $NH_3$ signal gradually fades into the

noise when the sampling deteriorates, resulting in an SNR decrease from 8.0 to 2.1 and a $\sigma_{abs}$ increase from $2.2 \times 10^{16}$ to $10.3 \times 10^{16}$ molec.cm$^{-2}$. By means of illustration, the insets of Figure 9 show the average of the 500 spectra measured inside the plume and used to calculate the SNR (blue) at different spectral resolutions. In red, a scaled Jacobian is shown with the signature of $NH_3$, illustrating that while the contribution of $NH_3$ to the average is easily seen at fine resolution, it becomes increasingly hard to identify $NH_3$ at coarser spectral resolution. As before, the FARs are all close to the theoretical value of

1.5% for the given threshold of MF. Yet, there are certain small features that are apparent in MF distributions, especially at coarser resolution. One of these is indicated with an arrow in Figure 9. The associated spectra have a large positive slope in the baseline between 900–1000 $cm^{-1}$. Inspection of visual imagery shows that this particular feature is related to a large rooftop. So even though the overall FAR statistics are favorable, localized false detections related to surface features are to be expected at small spatial scales.

The results related to the spectra measured over the Po Valley show the same tendencies, as summarized in the third column of Figure 3. Overall, these statistics are again fully consistent with those from IASI, when taking into account the larger noise of the Hyper-Cam measurements ($4.43 \times 10^{-4}$ W.(m$^2$.sr.cm$^{-1})^{-1}$, Figure A1) compared to that of the IASI instrument averaged in blocks of 1 $cm^{-1}$ ($1.73 \times 10^{-4}$ W.(m$^2$.sr.cm$^{-1})^{-1}$).





### 4.3.2 Well-chosen bands

As with the IASI spectra, we also degraded the Hyper-Cam spectra to just a few bands. The bands were selected as close as possible to the optimal ones found for IASI. The results are summarized in Figure 3. Generally, we observe that all metrics are remarkably close to each other for both campaigns, demonstrating that these are largely scene independent. Taking into account the larger noise of the Hyper-Cam measurements, the statistics follow closely those from IASI. In particular, we observe the general trend that the SNR and $\sigma_{abs}$ improve when the channels become narrower or when the number of channels increases.

The only exception is for the case of 4 bands of $10.4\,\mathrm{cm}^{-1}$, which performs slightly better than the 5 bands of $10.4\,\mathrm{cm}^{-1}$ over the Po Valley. This is likely related to the slight shifts in the positions of the bands. The FARs are all around the theoretical value of 1.5%.

We also observe again that configurations with few bands at higher resolution can perform just as well as a uniformly sampled instrument, supporting again the possibility of drastically reducing the number of spectral measurements to monitor $NH_3$. For 370 instance, for the Rosignano measurements, we see that the SNR and the $\sigma_{abs}$ of the 5-band $1.3\,\mathrm{cm}^{-1}$ configuration are similar, and even slightly better than the 40 continuous channels of $2.6\,\mathrm{cm}^{-1}$. Both distributions are compared in Figure 9b and 9e and indeed have a visually similar appearance. However, the FAR, at 1.6%, is slightly larger for the 5-band configuration, and there are also a few additional surface artifacts in the distribution with fewer channels. Somewhat surprisingly, we observe that generally the FAR for the campaign data does not vary as much as for IASI. This is likely due to the absence of sandy surfaces 375 with large and variable surface emissivity variations, posing particular difficulties for the MFs.

Figure 10 illustrates MF distributions obtained from spectra measured over the Po Valley sampled at $1.3\,\mathrm{cm}^{-1}$ and computed as continuous (panel (a)) and individual well-defined bands (panels (b) to (d)). The gradual increase in noise is visible, but the main $NH_3$ hotspots are clearly identifiable in all distributions, as well as the broad gradients across the map, confirming that $NH_3$ can be effectively observed with just a few channels, also at small spatial scales.

## 5 Conclusions

One of the crucial challenges in designing a satellite sounder is balancing the quality of the observations with the complexity and cost of a mission. For measuring atmospheric compounds, the key observational requirements are on the spatial resolution and the retrieval uncertainty that are intricately linked with the spectral range, resolution and radiometric noise of the instrument. Current sounders capable of measuring $NH_3$ offer excellent spectral and radiometric qualities, but lack the spatial 385 resolution for quantifying its many small-scale emission sources. While a hyperspatial high-resolution infrared sounder is technologically feasible, it has until now not yet been investigated whether the high spectral resolution offered by current sounders is required for effectively observing $NH_3$. In this study, we explored the feasibility of measuring $NH_3$ with instruments having a reduced spectral coverage and resolution compared to current and planned operational sounders. This objective is driven by the urgent need for effective monitoring of atmospheric $NH_3$ at high spatial resolution, especially in the context of excessive 390 agricultural and industrial emissions that significantly affect air quality and the environment.



A first general conclusion is that the 900–1000 $\mathrm{cm}^{-1}$ (or even the 925–975 $\mathrm{cm}^{-1}$) range is by far the most sensitive spectral range for infrared measurements of $NH_3$, and that from a pure retrieval uncertainty perspective, there is no large benefit of extending beyond. When it comes to instruments providing full spectral coverage with uniformly spaced channels, it was found that even a spectral resolution of 10 $\mathrm{cm}^{-1}$ can be adequate for monitoring $NH_3$, depending on the radiometric noise and the targeted retrieval uncertainty. However at such a resolution, spectral interferences start becoming increasingly important in the retrieval uncertainty budget, limiting inherently the retrieval to high columns only. For a general purpose $NH_3$ sounder of low-resolution and uniform sampling, we therefore would recommend a spectral resolution finer than 5 $\mathrm{cm}^{-1}$, noting that it is more efficient to observe $NH_3$ using fewer bands at finer sampling rather than more bands at coarser sampling.

The possibility of using a few dedicated broadband channels for measuring atmospheric gases has been demonstrated before. Varon et al. (2021) for instance retrieved methane ($CH_4$) column concentrations from two 10 nm-wide bands in the shortwave infrared of the Sentinel-2 MultiSpectral Instrument. We have shown that especially for sounders with a resolution of 5–10 $\mathrm{cm}^{-1}$, the location of the spectral bands is key, and if the instrumental design permits, 3–5 carefully chosen located bands can provide significant benefits on the retrieval uncertainty. For instruments with higher resolution (1–2 $\mathrm{cm}^{-1}$), already 2 or 3 carefully chosen microchannels can faithfully reproduce $NH_3$ distributions and outperform instruments with broader spectral bands.

Further validation was conducted using airborne Hyper-Cam measurements. The analysis of these spectra, measured at fine spatial resolution, confirmed the results obtained with IASI and demonstrated the applicability of a sounder with reduced spectral resolution for measuring $NH_3$ at small spatial scales, whether from industry or agriculture.

While it is clear that high spectral resolution offers significant benefits, it should be emphasized that all the results reported here assume uniform noise, which in practice will not be the case when comparing different engineered designs. For example, it is generally much easier to achieve a required noise level at low spectral resolution than at high. If a medium-low spectral resolution instrument were to be designed, the analysis should carefully take into account the estimated instrumental noise budgets that each of the potential designs could offer. The methodology developed and presented in this paper allows this, and can be easily generalized to variable noise levels. Indeed, for the case of uniform noise presented here, there are a plethora of well-chosen bands that theoretically have a very similar performance. Some of these alternative options are likely to perform better with a specific non-uniform noise profile.

## Appendix A: Downgrading spectra and total noise estimate

In this study, spectra were downgraded by averaging neighboring channels in blocks of different sizes $n$. For IASI, values of $n = 4, 8, 20$ and $40$ were used, resulting in spectra sampled at 1, 2, 5 and 10 $\mathrm{cm}^{-1}$. Block-averaging a spectrum is equivalent to convolving the original spectrum with a normalized boxcar function of width $n$, and selecting 1 out of $n$ channels of the resulting spectrum. The spectral resolution of this spectrum can be estimated from the full width at half maximum (FWHM) of the instrumental line shape of IASI (approximately Gaussian with a FWHM of 0.5 $\mathrm{cm}^{-1}$) convolved with this boxcar. Numerically, it was found to be very close to the sampling rate; see Table A1.





**Table A1.** The first part of the table compares the sampling and the equivalent spectral resolution from original IASI measurements and downsampled spectra. The second part lists the median of $\sigma$ between 900 and 1000 $\mathrm{cm}^{-1}$ for both diagonal and off-diagonal elements (in $\mathrm{W.(m^2.sr.cm^{-1})^{-1}}$). The bottom part shows the median values of the unapodized noise, the added noise and the total noise, provided in radiance units and converted in brightness temperatures for a reference temperature of 288 K.

| | Original IASI | Downgraded IASI measurements | | | |
|---|---|---|---|---|---|
| Sampling | $0.25\ \mathrm{cm}^{-1}$ | $1\ \mathrm{cm}^{-1}$ | $2\ \mathrm{cm}^{-1}$ | $5\ \mathrm{cm}^{-1}$ | $10\ \mathrm{cm}^{-1}$ |
| Resolution | $0.5\ \mathrm{cm}^{-1}$ | $1.01\ \mathrm{cm}^{-1}$ | $2.00\ \mathrm{cm}^{-1}$ | $5.00\ \mathrm{cm}^{-1}$ | $10.0\ \mathrm{cm}^{-1}$ |
| $\sigma_{ii}$ | $2.16 \times 10^{-4}$ | $1.61 \times 10^{-4}$ | $1.22 \times 10^{-4}$ | $0.77 \times 10^{-4}$ | $0.53 \times 10^{-4}$ |
| $\sigma_{ii+1}$ | $1.79 \times 10^{-4}$ | $0.46 \times 10^{-4}$ | $0.06 \times 10^{-4}$ | $0$ | $0$ |
| $\sigma_{ii+2}$ | $0.97 \times 10^{-4}$ | $0$ | $0$ | $0$ | $0$ |
| Unapodized noise | $3.46 \times 10^{-4}$ | $1.73 \times 10^{-4}$ | $1.23 \times 10^{-4}$ | $0.78 \times 10^{-4}$ | $0.56 \times 10^{-4}$ |
| Added noise | $/$ | $0$ | $1.22 \times 10^{-4}$ | $1.55 \times 10^{-4}$ | $1.64 \times 10^{-4}$ |
| Total noise $\mathrm{(W.(m^2.sr.cm^{-1})^{-1})}$ | $3.46 \times 10^{-4}$ | $1.73 \times 10^{-4}$ | $1.73 \times 10^{-4}$ | $1.73 \times 10^{-4}$ | $1.73 \times 10^{-4}$ |
| Total noise (mK) | 240 | 120 | 120 | 120 | 120 |

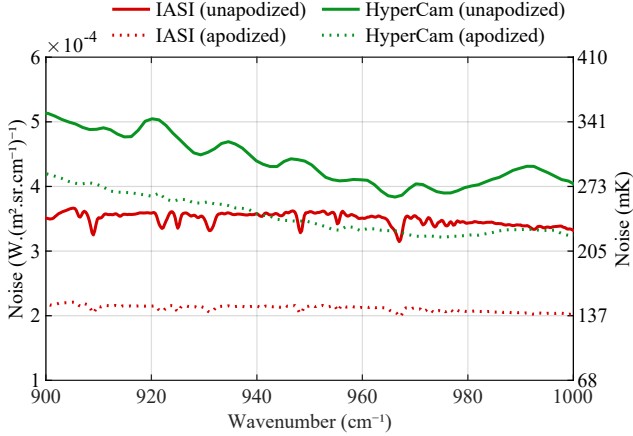

**Figure A1.** Radiometric noise estimated from native IASI and two by two averaged Hyper-Cam measurements between 900 and 1000 $\mathrm{cm}^{-1}$.

To compare these hypothetical instruments fairly, it is important to equalize their noise levels, and so Gaussian noise was
added to spectra to match the noise level of the spectra with the smallest sampling, as these have the largest noise. Noise covariance matrices were estimated from a principal component analysis (Atkinson et al., 2010; Serio et al., 2018). For IASI, as its spectra are quite strongly apodized, these matrices are found to have non-negligible off-diagonal elements, indicating correlated noise across nearby channels (Serio et al., 2014). Reversible apodization does not affect the total noise in the





spectrum, but does make it more difficult to compare noise levels. Below, we show that the uncorrelated noise, i.e. the diagonal
noise before apodization, can be recovered by simply summing up the columns of the correlated noise covariance matrix.

Table A1 summarizes, for both original and downgraded IASI spectra, the apodized noise (diagonal and off-diagonal ele-
ments of the noise covariance matrix), and total unapodized noise. It also tabulates for each choice of $n$ how much Gaussian
noise was added and the resulting total noise, both in radiance units and brightness temperatures. Figure A1 shows the original
IASI apodized and unapodized noise between 900 and 1000 $\mathrm{cm}^{-1}$.

The same procedure was applied to the Hyper-Cam data. Spectra were downgraded by block-averaging neighboring channels
($n = 1, 2, 4$ and $8$), resulting in spectra sampled at 1.3, 2.6, 5.2 and 10.4 $\mathrm{cm}^{-1}$. Gaussian noise was added to these spectra to
match the unconvoluted noise level of the original measurements, estimated at around 300 $\mathrm{mK}$ from the noise covariance
matrix. The apodized and unapodized noise calculated from the original data measured over Rosignano are shown in Figure
A1 (in green).

We end this section with a proof of the relation between the apodized and unapodized noise covariance matrix. Let $\mathbf{S}_\epsilon$
be a diagonal noise covariance, with $\boldsymbol{\sigma}$ the square root of its diagonal. We now investigate how this covariance matrix is
altered by convolution or apodization. We write the convolution as $w_{-n}, w_{-n+1}, \dots, w_{n-1}, w_n$, with $\sum_k w_k = 1$. We assume
the convolution only spreads a few elements wide (much smaller than the length of the spectrum). The convolution has the
following effect on a given spectral noise $\boldsymbol{\epsilon}$

$$\tilde{\epsilon}_j = \sum_k w_{k-j} \epsilon_k.$$

This can be written in matrix notation as

$$\tilde{\boldsymbol{\epsilon}} = \boldsymbol{w} \star \boldsymbol{\epsilon} = \mathbf{W}\boldsymbol{\epsilon},$$

with $W_{jk} = w_{k-j}$. Using the propagation of uncertainty formula, the noise covariance matrix of the convoluted spectra takes
the form (Tellinghuisen, 2001)

$$\tilde{\mathbf{S}}_\epsilon = \mathbf{W}\mathbf{S}_\epsilon \mathbf{W}^{\mathrm{T}},$$

or

$$\tilde{\mathbf{S}}_{\epsilon,ij} = \sum_{kl} \mathbf{W}_{ik} \mathbf{S}_{\epsilon,kl} \mathbf{W}_{jl} = \sum_k \mathbf{W}_{ik} \sigma_k^2 \mathbf{W}_{jk},$$

where we have used the fact that $\mathbf{S}_\epsilon$ is assumed diagonal. This equation gives the relation between the apodized and unapodized
covariance matrix. Let us now evaluate the vector obtained by summing the rows of $\tilde{\mathbf{S}}_{\epsilon,ij}$

$$\sum_i \tilde{\mathbf{S}}_{\epsilon,ij} = \sum_{ik} \mathbf{W}_{ik} \sigma_k^2 \mathbf{W}_{jk} = \sum_k \left( \sigma_k^2 \mathbf{W}_{jk} \sum_i \mathbf{W}_{ik} \right).$$

Using the fact that the sum of all rows of $\mathbf{W}$ equals an all-one vector (disregarding edge effects), we find

$$\sum_i \tilde{\mathbf{S}}_{\epsilon,ij} = \sum_k \sigma_k^2 \mathbf{W}_{jk}.$$

Assuming a small kernel, slowly varying noise ($\sigma_k \approx \sigma_j$), and again away from end points, we find that

$$\sum_i \tilde{\mathbf{S}}_{\epsilon,ij} = \sigma_j^2.$$



We can therefore recover the unconvoluted noise $\sigma^2$ by simply adding up the columns of the noise covariance matrix.

*Author contributions.*    LN and LC conceptualized the research, analyzed and interpreted the results, prepared the figures and wrote the first version of the manuscript. TR organized the campaigns and carried out the measurements. All authors took part in the campaigns planning, in the discussions and revisions of the manuscript.

*Competing interests.*    One of the (co-)authors is a member of the editorial board of Atmospheric Measurement Techniques. The authors have
no other competing interests to declare.

*Acknowledgements.*    The authors acknowledge the European Space Agency (ESA) for funding the airborne campaigns. The authors also gratefully acknowledge GFZ German Research Centre for Geosciences and Freie Universität Berlin (FUB) for the use of the Hyper-Cam instrument and for their support and cooperation during the campaign. This publication is supported by the French Community of Belgium in the framework of a FRIA grant. L. Clarisse is Senior Research Associate supported by the Belgian F.R.S.-FNRS. The authors thank Tim
Hultberg and Jean Vander Auwera for discussions on principal component analysis and spectral resolution.



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
