# Peer review of "Towards a low-resolution infrared sounder for monitoring atmospheric ammonia (NH3) at high spatial resolution"

_EGUsphere, 2024_

## Referee Comment (RC1)

"Towards a low-resolution infrared sounder for monitoring atmospheric ammonia (NH3) at high spatial resolution"

Lara Noppen, Lieven Clarisse, Frederik Tack, Thomas Ruhtz, Martin Van Damme, Michel Van Roozendael, Dirk Schuettemeyer, and Pierre Coheur

The study detailed in this paper describes an evaluation tool to investigate the optimum configuration for a dedicated satellite based NH3 sounding instrument. A modelled spectral simulation, using line-by-line radiative simulation, is used to identify a selection of optimal configurations for a hypothetical spaceborne imager capable of high spatial resolution measurements with sufficient NH3 retrieval sensitivity to identify and isolate single point NH3 emission sources.

The authors apply these configurations to existing data sets acquired by IASI and Hyper cams to compare and contrast NH3 identification over a large geographical range for IASI and for selected regional sites in the case of Hyper-cams.

The observational data sets from the IASI instrument on METOP-B and the Hyper-Cam LW system on a Cessna T207A aircraft are degraded to simulate measurements from low resolution and descoped spectral range instruments. System noise, background "clutter", signal-to-noise and False alarm rates are defined and used to assess NH3 column retrieval capability as the observed spectral resolution is reduced over a continuous spectral range and/or a combination of selective bands within a given spectral range are isolated.

The authors demonstrate that low-cost descoped instruments, with sufficient retrieval sensitivity for NH3 emissions monitoring can be achieved.

The paper is very well written and structured. The description of methodology, application and results are clear and the assessment and conclusion sound. I believe the evaluation tool offers a new and solid base to undertake future instrument optimisation.

I recommend this paper for publication with some minor modifications.

The overall paper can be a little improved if the authors could discuss the current state-of-the-art to put this tool into a real-world perspective, for instance some detail on the available narrow-band filters/detector combinations that might be required to isolate spectral bands, and their efficiency would be informative. Will these form part of an FTS system, to provide the spectrally resolved measurements after isolating individual bands?

The authors indicate that only the 9:30 local time IASI data were used for this study. Is this to better match a daytime peak in NH3 emissions? was the 21:30 overpass ever considered to evaluate the capability to observe a diurnal cycle in emission or does the reduction in thermal

contrast make this unfeasible. Emissions in April 2020 would have been impacted by the COVID lockdown, given these may therefore not be typical emission scenarios, i.e. potential for reduced background levels, are there any implications for the comparison.

**4.1 from simulated spectra**

Reading through the description on the measurement uncertainty as a function of range and resolution, figure 1b indicates to me that, for the two ranges indicated, the resolution should not be higher than 1.21 cm-1 for the given NH3 retrieval baseline at the specified noise(temperature) level. I think this can be more explicitly stated. Hyper-Cam is then shown to be on the edge of this limit and gives a good justification for the pixel averaging used later.

line 208: missing units on the 2 (cm)

**4.2 From IASI**

This comparison of simulated instrument configurations is very informative. If IASI at its native resolution is to be taken as "truth", as shown in figure 2, reducing the number of bands and moving towards courser spectral resolution leads to a smoothing of the variance in geographical NH3 column, this is seen in a relative enhancement of NH3 over Eastern Europe near Hungary. Taking this in context of not just reproducing NH3 distribution but also monitoring of emissions and emission levels, these results indicate that a minimum of 4 channels targeting both major NH3 absorption bands at a resolution of 2 cm-1 would be a suitable configuration.

**4.3 From Hyper-Cam**

This section extends that undertaken in section 4.2 to Hyper-Cam measurements and shows essentially similar behaviour in sensitivity to variable spectral resolution and limited band number, with some interesting outliers associated with higher scene resolution, as seen over the resolved rooftop.

Would such a fine spatial scale from space be feasible? If the hypothetical instrument, this study provides tools to evaluate, were based on a Hyper-Cam in low Earth orbit at 600 km, the single pixel spatial scale increases from 5 m x 5 m to 100 m x 100 m, with no equivalent improvement in spectral noise. It might be insightful to average the Hyper-Cam pixel measurements to simulate the spatial resolution of measurements from low Earth orbit and re-run the sensitivity study. Alternatively, can the authors provide an explanation of how this might impact the behaviour in MF, σ(abs), SNR and FAR shown in the study using the Hyper-Cam.

I would like to see a little more detail on how the Hyper-Cam measurements shown were obtained, the timeframe and flight tracks.

---

## Author Response (AR1)

**Reviewer #1**

Thank you very much for your positive assessment of the manuscript and your comments, which have been addressed as detailed below (in blue).

The study detailed in this paper describes an evaluation tool to investigate the optimum configuration for a dedicated satellite based NH3 sounding instrument. A modelled spectral simulation, using line-by-line radiative simulation, is used to identify a selection of optimal configurations for a hypothetical spaceborne imager capable of high spatial resolution measurements with sufficient NH3 retrieval sensitivity to identify and isolate single point NH3 emission sources.

The authors apply these configurations to existing data sets acquired by IASI and Hyper cams to compare and contrast NH3 identification over a large geographical range for IASI and for selected regional sites in the case of Hyper-cams.

The observational data sets from the IASI instrument on METOP-B and the Hyper-Cam LW system on a Cessna T207A aircraft are degraded to simulate measurements from low resolution and descoped spectral range instruments. System noise, background "clutter", signal-to-noise and False alarm rates are defined and used to assess NH3 column retrieval capability as the observed spectral resolution is reduced over a continuous spectral range and/or a combination of selective bands within a given spectral range are isolated.

The authors demonstrate that low-cost descoped instruments, with sufficient retrieval sensitivity for NH3 emissions monitoring can be achieved.

The paper is very well written and structured. The description of methodology, application and results are clear and the assessment and conclusion sound. I believe the evaluation tool offers a new and solid base to undertake future instrument optimisation.

I recommend this paper for publication with some minor modifications.

The overall paper can be a little improved if the authors could discuss the current state-of-the-art to put this tool into a real-world perspective, for instance some detail on the available narrow-band filters/detector combinations that might be required to isolate spectral bands, and their efficiency would be informative. Will these form part of an FTS system, to provide the spectrally resolved measurements after isolating individual bands?

We thank the reviewer for this valuable suggestion, but our study focuses on evaluating theoretical performance rather than specific instrument implementations (which are also outside our domain of expertise). However, we agree that this is an important consideration for future work.

The authors indicate that only the 9:30 local time IASI data were used for this study. Is this to better match a daytime peak in NH3 emissions? was the 21:30 overpass ever considered to evaluate the capability to observe a diurnal cycle in emission or does the reduction in thermal contrast make this unfeasible. Emissions in April 2020 would have been impacted by the COVID

lockdown, given these may therefore not be typical emission scenarios, i.e. potential for reduced background levels, are there any implications for the comparison.

We chose to use only the morning measurements because the thermal contrast is stronger during the day, which is essential for infrared NH₃ monitoring. Since our study focuses on the design of a dedicated NH₃ sounder, we assume it would observe the Earth under optimal thermal contrast conditions, ideally between 11:00 and 14:00. It was therefore natural to select the morning IASI data for our analysis.
We chose a day in April 2020 without worrying about the covid lockdown, as NH₃ emissions are predominantly due to agricultural activities, which were largely unaffected by lockdown measures. Moreover, our study is based on relative comparison between original and downsampled data, rather than absolute concentration levels. Therefore, any potential reduction in the background level does not affect our results.

**4.1 From simulated spectra**

Reading through the description on the measurement uncertainty as a function of range and resolution, figure 1b indicates to me that, for the two ranges indicated, the resolution should not be higher than 1.21 cm-1 for the given NH3 retrieval baseline at the specified noise(temperature) level. I think this can be more explicitly stated. Hyper-Cam is then shown to be on the edge of this limit and gives a good justification for the pixel averaging used later.

In Figure 1b, the black curves (left axis) show the measurement uncertainty as a function of spectral sampling for a fixed noise level (100 mK), while the orange curves (right axis) show the required noise level to achieve a fixed uncertainty ($3 \times 10^{15}$ molec.cm$^{-2}$). The two axes are independent, and so the crossing point of both curves is arbitrary.
With this figure, we understand that the finer the spectral resolution, the higher the instrumental noise can be to achieve the same measurement uncertainty (orange); and vice versa that for a given instrumental noise, that coarser spectral resolution gives rise to a larger measurement uncertainty (black). This simulation shows that if we do not take into account uncertainties due to other parameters (which is done in the next sections), any measurement uncertainty can be achieved at any spectral resolution, provided the instrumental noise is sufficiently low. This is also expressed in Equation 11.

line 208: missing units on the 2 (cm)
It is now corrected, thank you.

**4.2 From IASI**

This comparison of simulated instrument configurations is very informative. If IASI at its native resolution is to be taken as "truth", as shown in figure 2, reducing the number of bands and moving towards courser spectral resolution leads to a smoothing of the variance in geographical NH3 column, this is seen in a relative enhancement of NH3 over Eastern Europe near Hungary. Taking this in context of not just reproducing NH3 distribution but also monitoring of emissions and emission levels, these results indicate that a minimum of 4 channels targeting both major NH3 absorption bands at a resolution of 2 cm-1 would be a suitable configuration

We think that the relatively higher $NH_3$ levels observed over Hungary in Figure 4d are artefacts likely due to the random noise that was applied. We agree that the 4 channels of 2 $cm^{-1}$ yield an $NH_3$ distribution that closely matches the original IASI distribution, however as pointed out in the text, 3 well-chosen bands of 1 $cm^{-1}$ are associated with even better performance metrics (SNR, FAR and $\sigma_{abs}$).

**4.3    From Hyper-Cam**

This section extends that undertaken in section 4.2 to Hyper-Cam measurements and shows essentially similar behaviour in sensitivity to variable spectral resolution and limited band number, with some interesting outliers associated with higher scene resolution, as seen over the resolved rooftop.

Would such a fine spatial scale from space be feasible? If the hypothetical instrument, this study provides tools to evaluate, were based on a Hyper-Cam in low Earth orbit at 600 km, the single pixel spatial scale increases from 5 m x 5 m to 100 m x 100 m, with no equivalent improvement in spectral noise. It might be insightful to average the Hyper-Cam pixel measurements to simulate the spatial resolution of measurements from low Earth orbit and re-run the sensitivity study. Alternatively, can the authors provide an explanation of how this might impact the behaviour in MF, σ(abs), SNR and FAR shown in the study using the Hyper-Cam.

Within the phase 0 study supporting the Nitrosat project, it was shown that a spatial resolution of 500 m x 500 m is a realistic target with current technologies. This is not the case for a resolution as fine as 100 m x 100 m.

Downgrading Hyper-Cam at such a resolution may not be representative of what would be observed from space, as satellite instruments would differ in technology, and benefit from a more stable platform. The analysis presented in Section 4.2 from IASI measurements provides a more realistic view of what can be expected from spaceborne measurements. The Hyper-Cam analysis complements this by showing that spatial resolution does not impact the drawn conclusions.

I would like to see a little more detail on how the Hyper-Cam measurements shown were obtained, the timeframe and flight tracks.

Following this comment, we added the starting time of some flight tracks on Figures 7 and 8 (see updated figures below), allowing better to distinguish the different flight tracks and directions of flight, and complementing the brief description of Hyper-Cam measurements available in Section 2 of the manuscript.

[Figure]

Updated figure 7

[Figure]

Updated figure 8

**Reviewer #2**

Thank you very much for your assessment of the manuscript. We have addressed your comments as detailed below (in blue).

The paper covers the potential design of a more ammonia specific sensor.  Presently there are no satellites in orbit specifically designed to measure ammonia.  Current satellites like IASI and CrIS are meteorological satellite sensors (designed to measure things like temperature and water vapor) that have been used to also monitor ammonia due to their large spectral range.  Thus, these sensor design trade studies looking at specifically monitoring ammonia (high spatial resolution) are important and timely to help design future ammonia monitoring instruments.  The paper is well written, and the results are provided in logical order.   As with any trade study, the scope of what can be evaluated is large.   The following are a few general comments for the authors to address.

1. The authors select only one spring day of April $8^{th}$ in 2020. IASI has a large range of observations over the years that span many atmospheric conditions.   One would not want to design an instrument to monitor only very small amounts (like in the non-growing seasons), but it would be good to provide the reader with some idea of the conditions being covered on this day.   For example, are they represented of what an instrument would on average see when monitoring, or are they under more ideal springtime conditions (e.g. histogram of the ammonia and atmospheric states from this day vs more typical IASI global observations, etc.).   It is true when comparing sensor designs, they are all using the same inputs, so the relative comparison is fine, but it would be good to give the reader a better sense if the results are from more favourable or typical remote sensing conditions.  Were any data filtered on this day?

   The IASI data from April $8^{th}$ in 2020 were indeed selected because they feature large $NH_3$ plumes, allowing us to clearly illustrate the study. However, the metrics were chosen to be largely independent of the specific scene. Following your comment, and to illustrate this, we extended our analysis based on IASI data measured 6 months later, in autumn, when $NH_3$ levels are typically lower and infrared observation conditions are less optimal. As expected, the performance metrics are in very good agreement with those derived from spring observations.

   To reflect this in the manuscript, we have added a paragraph at the end of Section 4.2 and included the results in a new Appendix B.

   Section 4.2
   *Finally, to demonstrate the robustness of these findings, we repeated the same set of tests using IASI measurements acquired six months later, on October 8, 2020. In autumn, $NH_3$ observation conditions are less favourable, resulting in generally lower MFs. However, since our analysis focuses on relative differences compared to the original IASI measurements, the*

*performance metrics derived from downgraded data are in very good agreement with those obtained from the spring dataset (see Appendix B).*

*Appendix B*
*The tests described in Section 4.2 were also applied to IASI data collected on October 8, 2020, to assess the robustness of our conclusions under less favourable observation conditions. Figure B1 presents the original MF distribution, illustrating that the overall MF levels are lower than in April. Following the same methodology as for the spring dataset, we evaluated the $NH_3$ detection with the four performance metrics introduced in Section 4.2. To ensure that the SNR of the initial distribution is comparable to the SNR calculated from the spring dataset, the mean SNR was calculated from a tuned set of spectra, i.e. as the average MF for spectra with $MF_{IASI} > 14.2$. The FAR was calculated as before as the fraction of observations for which MF > 2.5 inside the "out" boxes defined in Figure B1. The values $\sigma_{abs}$ and $\sigma_{noise}$ were calculated as before with Equations 3 and 6, with $(S_g, y_g)$ constructed from all spectra with $MF_{IASI} < 1.5$.*
*The measurements were then downgraded as previously described and assessed using these performance metrics. Figures B2, B3 and B4 summarise the results, showing that they are almost identical to those derived from the spring dataset. The largest differences are observed in the FARs, which is expected due to random nature of these. Overall, these results confirm the consistency of the conclusions drawn from spring measurements and highlight the robustness of the performance metrics.*

[Figure]

*Figure B1. $MF_{IASI}$ distribution calculated from the morning IASI overpass on October 8, 2020 from Clarisse et al. (2023). The signal-to-noise ratio (SNR), the false alarm rate (FAR) and the uncertainty ($\sigma_{abs}$) are indicated on top. The "out" boxes correspond to the areas used to estimate the FAR.*

[Figure]

Figure B2. (a) SNR, (b) FAR and (c) $\sigma_{abs}$ as a function of the number of channels for the degraded IASI measurements (October 8, 2020). The triangles show the results for continuous channels between 900 and 1000 cm$^{-1}$, while the dots represent for the well-chosen spectral bands of different widths, as indicated by their respective color. The open triangles on $\sigma_{abs}$ plots indicate $\sigma_{noise}$.

[Figure]

[Figure]

2. Related a bit to the first comment. Table 2 contains the resulting SNR values for a variety of band selections. What seems somewhat surprising is the magnitude of the overall SNR values are higher than expect, especially for the lower spectral resolutions. The SNR can be defined in many ways, so that might be a part of it, also the instrumental spectral noise level is not high. However, related to comment (1), it does make me wonder if the remote sensing conditions are more favourable with there not being on average as many conditions in the

test dataset that would produce an ammonia radiance signal that is close or below the detection limit of the sensor.

We agree that the SNRs depend fully on the selected spectra for the calculation. For the April case we selected observations for which $MF_{IASI}>20$, explaining the high SNR overall. However, the key points in our analysis are not the absolute values of the SNR, but the difference of SNRs between different configurations. As shown in the additional analysis dealing with autumn IASI data described above, the SNRs (and other performance metrics) remain consistent even under less favorable conditions.

3. The spectral bands selected and results for a give sensor design will depend on the atmospheric conditions used in the trade study. Did the authors look at the changes in the results depending on the atmospheric conditions (e.g. ideal vs typical vs challenging remote sensing conditions)?

As noted in the conclusion, in the case of this study assuming a uniform noise, there exist a multitude of well-chosen bands combinations that theoretically have a very similar performance. This flexibility suggests that the optimal band selection is relatively stable across varying atmospheric conditions.
To confirm this, we re-ran the algorithm defining the two optimal bands minimizing $\sigma_{abs}$ (as explained in Section 4.2.2) using IASI data measured on October 8, 2020. The resulting optimal bands differ slightly from that obtained from the spring dataset, but the resulting performance metrics for $NH_3$ detection are very similar, as illustrated with the following table.

| | $1\ cm^{-1}$ | $2\ cm^{-1}$ | $5\ cm^{-1}$ | $10\ cm^{-1}$ |
|---|---|---|---|---|
| $\sigma_{abs}$ from the initial 2-band combination (derived from the spring dataset) | $2.1 \times 10^{16}$ | $2.1 \times 10^{16}$ | $4.0 \times 10^{16}$ | $6.0 \times 10^{16}$ |
| $\sigma_{abs}$ from the 2-band combination noted below (derived from the autumn dataset) | $2.0 \times 10^{16}$ | $2.1 \times 10^{16}$ | $3.6 \times 10^{16}$ | $6.0 \times 10^{16}$ |
| 2-band combination derived from the autumn dataset | 967.50 970.50 | 967.25 970.50 | 965.75 970.75 | 932.00 942.00 |

4. It would be good for the authors to provide comments in the paper on difference between this approach over a traditional "microwindow" selection based on information content.

Thank you for this suggestion, we have now added the following paragraph, discussing the differences:

*In terms of channel-selection methodology, our approach attempts to find the extremum over all channels combinations, whereas the traditional microwindow selection process as in (Rodgers et al., 1998) uses an iterative approach, where one channel is added in turn, each time adding a maximum of information. This process is computationally more efficient (and*

*thus suitable for a large number of channels) but is less likely to yield a global extremum than our approach.*

5.  In the spectral resolution trade study the authors appear to assume a constant instrument noise level. It has been shown in previous ammonia sensor design studies that often the instrument noise is the larger driver. The authors do mention in the conclusions that the analysis can be expanded in the future to consider different noise levels, which is good. When discussing overall results it is good to make it clear that they are for a specific noise level and that any specific sensor design will also depend on the instrument noise.

    As noted, the results presented in the study are indeed noise specific. This is already made explicit with the discussion around Figure 1, but we now explicitly mention this again in section 4.2.1 as follows:
    *It should be noted that the results presented here are specific to the assumed noise level, but the observed trends remain general, as further supported in the following section dealing with noisier data.*

6.  The spectral selection will depend on cross-state errors (e.g. temperature, water vapor, spectroscopic parameters, etc.) as noted and accounted for by the authors in their analysis, which is great. Accounting for the impact of these interfering species is only as good as the specified error estimates. It would be good if the authors could provide more information on the generation of the estimates. Also, since the authors can easily produce simulated retrievals, did they perform any Monte Carlo type statistical tests to see if the estimates are robust. For example, put in errors in the temperature, water vapor, etc. (e.g. ECMWF) on a pixel-by-pixel basis and see the impact. This will be particularly important for any sensor design that is not on a more traditional meteorological sensor and does not have coincident water vapor and temperature sounding.

    While we account for cross-state errors due to e.g. surface emissivity variations or water vapour, via the use of the generalized noise covariance matrix, we do not explicitly model the propagation of uncertainty errors in these parameters or perform a simultaneous fit of these parameters. In addition, one caveat of our analysis, as acknowledged at the bottom of section 3.1, is that we assume to have perfect knowledge of the Jacobian. Thermal contrast (and thus temperatures), spectroscopy, and unknowns related to the vertical distribution of $NH_3$ are thus not taken into account. As these are not instrument dependent (if temperatures are taken from an external source), these do not affect the conclusions of the work.
    While the presented methodology has a lot of advantages, it does not replace a full end-to-end simulation and retrieval that would allow the type of Monte Carlo tests that you suggest. We agree that for a future sensor development this would be required.